# Conversion of Spent Coffee and Donuts by Black Soldier Fly (*Hermetia illucens*) Larvae into Potential Resources for Animal and Plant Farming

**DOI:** 10.3390/insects12040332

**Published:** 2021-04-08

**Authors:** Hayden Fischer, Nicholas Romano, Amit Kumar Sinha

**Affiliations:** Center of Excellence in Aquaculture and Fisheries, University of Arkansas at Pine Bluff, 1200 North University Drive, Pine Bluff, AR 71601, USA; fischeh1948@uapb.edu (H.F.); sinhaa@uapb.edu (A.K.S.)

**Keywords:** insect farming, frass, NPK, spent coffee, prepupae

## Abstract

**Simple Summary:**

Unsustainable farming practices have depleted the quantity and quality of topsoil and, moreover, 30–40% of the food produced ends up in landfills. These issues can be simultaneously mitigated by producing black soldier fly (*Hermetia illucens*) larvae that convert food waste into two resources. This includes the black soldier fly (*Hermetia illucens*) larvae (BSFL) itself as a rich source of protein and lipid for animals as well as the “frass” which is the leftover organics that can be used as fertile compost. The aim of this study was to examine the applications of two widely available resources, spent coffee grounds and donut dough, as food for BSFL. The proximate, fatty acid and amino acid composition demonstrates that a blend of these resources produced BSFL of similar quality as soybean meal. Moreover, the left behind frass had a similar nutritional profile as many organic fertilizers. Therefore, BSFL farming yields potential resources for animal and plant farming from otherwise discarded waste.

**Abstract:**

Nutritionally unbalanced organic waste can be converted into potential resources for animal and plant farming by culturing black soldier fly (*Hermetia illucens*) larvae (BSFL) and prepupae (BSFP). BSFL and BSFP are rich sources of protein and lipids, while the leftover excrement called “frass” can be used as an organic fertilizer. Using readily available resources, BSFL were cultured on spent coffee, donut dough or an equal blend for 35 days. Survival, productivity, daily pupation and biochemical composition of BSFL and BSFP were measured along with the nitrogen-phosphorus-potassium values of the frass. Survival was highest in the blend compared (81%) to spent coffee (45%) or dough (24%); however, BSFL and BSFP were significantly longer and heavier from dough. Stage and food significantly influenced the protein, lipid and glycogen content of the BSFL and BSFP, which tended to be higher in the latter. While fatty acids were often significantly higher in BSFL fed spent coffee, the amino acid composition of BSFL was generally higher in dough. Frass from the blend had significantly highest nitrogen content, while potassium and phosphorus were significantly higher and lower from spent coffee, respectively. Although coffee and donut dough were suboptimal substrates for BSFL, a blend of these produced BSFL and frass that were nutritionally comparable to soybean meal and many organic fertilizers, respectively.

## 1. Introduction

Waste management is becoming a more challenging issue with a growing population where it is estimated that one-third of all the food produced is wasted [1]. Food waste contributes 15.2% (or 40 million tons) of the 267.8 million tons of municipal solid waste generated each year in the United States [2]. Only 2.7% of food waste is used for recycling/composting with the remainder ending up in landfills. Within landfills, food waste substantially contributes to greenhouse gas emissions and produces toxic leachate that can end up in waterways and attracts pests [3,4]. This situation will be compounded with an increasing global population that is estimated to reach 8.6 billion in another decade [5] leading to a solid waste production of over 2.2 billion tons [6]. 

A more sustainable approach to waste management can include the use of vermicomposting from insects, such as black soldier fly larvae (BSFL). BSFL can thrive on various food waste including fruits/vegetables, corn/rice straws, kitchen/restaurant food waste and various grains including distiller grains and wheat [7,8,9,10,11]. Depending on the type of food provided, this can greatly affect the nutritional value of BSFL [12,13,14,15]. 

At harvest, this process can yield two sustainable products; the BSFL as food for terrestrial and aquatic animals while the excrement of BSFL known as “frass” is a natural organic fertilizer with applications for plant farming. Depending on the type of provided food, BSFL can have a crude protein content up to 56% but as low as 13.2% [16], but the amino acid content of BSFL is relatively stable [15,16,17]. The consistency of the amino acid content in BSFL is beneficial, particularly because the composition is similar or even better than soybean meal that is often the dominant protein source in terrestrial and aquatic animal diets [18]. Moreover, there have been some positive findings when BSFL were included in the diets of terrestrial and aquatic animals. For example, the inclusion of BSFL as supplemental food to broiler and egg-laying hens was shown to significantly enhance muscle growth and egg production, respectively [19,20,21]. Recently, Kumar et al. [22] found that BSFL meal prevented intestinal enteritis associated with dietary excessive soybean meal in rainbow trout.

Findings that demonstrate the benefits of BSFL will likely increase the demand for this product, which will inevitably leave behind a substantial amount of frass. Perhaps to a greater extent than BSFL, the type of food provided will have a large impact on the frass composition [23]. For example, frass obtained from BSFL fed on distillers’ dried grains with solubles could be used as a viable fish ingredient [24], while those fed vegetable, fruits and other plants are more appropriately used as an organic fertilizer that can provide similar benefits to plant growth as synthetic fertilizers [25,26,27,28]. 

Some of the most commonly thrown away resources in the world include spent coffee grounds and bread/dough. It is estimated that up to 25 billion cups of coffee are ingested each day, leading to over 6 million tonnes of spent coffee grounds ending up in landfills each year [29]. Exact figures on the amount of thrown away bread and dough are difficult to find. However, in just UK households it is believed that 44% of bread is thrown away, which is worth over 13 million UK pounds [30]. Although the performance of BSFL fed spent coffee grounds have not yet been directly compared with other food types, it is believed spent coffee is not an ideal food for BSFL [31,32]. However, the growth and development of BSFL might be enhanced when using blends of readily digestible sources, such as dough. Indeed, based on comparing several streams of organic waste, Lalander et al. [16] found that food with more easily accessible carbon with a high protein content supported the best BSFL growth. Similarly, among various waste sources, kitchen waste led to the heaviest BSFL, which was suggested to be from highly digestible fat and caloric content [7].

The aim of this study was to compare the growth, development and nutritive value of BSFL fed spent coffee, dough or an equal blend of these ingredients as well as the nitrogen-phosphorus-potassium (NPK) values of the resulting frass.

## 2. Materials and Methods

### 2.1. Source of Insects and Indoor Room

The black soldier fly larvae used in this study were a third generation that were originally purchased from Josh’s Frogs, Owosso, MI. The adults were allowed total access in the room (6 m long × 6 m wide × 5.4 m high), which was temperature controlled at 88 °F (31 °C) at a relative humidity of around 45%. The flies had access to sugar water in petri dishes and each day the room was manually sprayed with water to maintain humidity and allow the flies to drink. 

### 2.2. Experimental Set Up and Design

A total of nine staked systems (0.9 m × 1.2 m × 1.5 m) were constructed out of steel and the details are described in Fischer and Romano [33], with slight modifications. Briefly, each system had three layers (0.9 m × 1.2 m top, 0.6 m × 0.9 m bottom, and 3.3 m height) giving a total space of 0.7 m^3^. Each layer had hardware cloth and window screening to hold the feed ingredients. Each layer was a trapezoid and the angles were provided to allow the larvae to self-harvest after reaching the prepupae stage. A metal trough that exceeded the length of each system was placed on each side to catch and collect the prepupae. Each trough had a ½ inch layer of sand to allow the larvae to burrow as well as to soften the fall. A tarp was placed on the top to provide some shade and a central slit was made to allow access for the adult flies.

There was a total of three food treatments of equal weight (6.8 kg), which included spent coffee grounds, an equal blend of coffee grounds and donut dough and donut dough. The spent coffee grounds and donut dough were donated from a local coffee and donut shop, respectively. These substrates were chosen based on being free and readily available in the area as well as having a different texture and biochemical composition. To make the blend, the dough was cut into small pieces and equally mixed with the spent coffee grounds. The proximate composition as well as the potassium (K) and phosphorus (P) of the spent coffee grounds and dough were measured at the Agricultural Experiment Station Chemical Laboratories (AESCL) at University of Missouri-Columbia according to AOAC [34] methods (Table 1). The amino acid and fatty acid composition of the spent coffee, dough and blend were also measured at AESCL, University of Missouri-Columbia, according to AOAC [34] methods 982.30 (a,b) and 996.06, respectively (Table 2 and Table 3).

Prior to adding the substrates, equal amounts of water (500 mL) were added to provide a moist texture. After adding the water to each layer, approximately 500 larvae (average initial larval weight = 25 mg) were equally distributed among the three layers. A mister system was set up to provide a spray of water to each layer for approximately 30 s each day. No additional substrate was provided throughout the study. Throughout the study, the trough was observed for any prepupae twice daily (morning and afternoon), and when found, they were immediately placed in a small plastic bag, labelled for the day/treatment and then placed in a −20 °C freezer for later analysis.

### 2.3. Sampling and Biochemical Analysis

After 35 days, the hardware cloth was removed from each layer, and the contents (frass and larvae) were emptied onto a tarp. All the larvae were counted and differentiated between larvae or prepupae. These were then thoroughly cleaned under running tap water, blotted dry, placed in air-tight plastic bags and kept at −20 °C. Within a week, all the larvae and prepupae were weighed to determine the overall gross and net productivity, using the following calculations.
Gross production = final weight/time/area of composter (0.7 m^3^)
Net production = [final weight − initial weight]/time/area of composter (0.7 m^3^)

From a subpopulation of 50 from each stage, these were measured for their moisture, protein, lipid and glycogen content, while only the larvae were measured for the amino acid and fatty acid composition (Section 2.4). From the remaining frass, this was collected, weighed and analysed for the moisture as well as nitrogen (N), P, K and calcium (Ca) content (Section 2.5).

### 2.4. Biochemical Analysis of the Larvae

The larvae and prepupae were cut into small pieces and dried in the oven at 60 °C until constant weight. The moisture was calculated as the percentage of weight lost after drying. After drying, the larvae and prepupae were further homogenized separately into a fine powder and measured for protein content by Bradford’s method [35] using bovine serum albumin as standard. Glycogen content was measured using anthron reagent and a glycogen standard [36]. Total lipid was extracted by methanol chloroform and measured with a tripalmitin standard following method of Bligh and Dyer [37]. The amino acids and fatty acids composition of only the larvae were measured at AESCL at University of Missouri-Columbia and were analysed according to AOAC [34] methods 982.30 and 996.06, respectively.

### 2.5. Frass Composition

The remaining frass in each replicate were completely removed, weighed (0.01 g) and then dried to constant weight in an oven at 70 °C. The NPK content as well as the Ca content of the frass were measured at AESCL according to AOAC [34] methods (method 985.01) using an inductively coupled plasma-optical emission spectroscopy.

### 2.6. Statistical Analysis

All data were subjected to a one-way ANOVA after prior confirmation of homogeneity of variance and normality. If significant differences were detected (*p* < 0.05), a Duncan’s post-hoc test was performed to identify differences among the treatments. When comparing the lengths and weights as well as the moisture, protein, lipid and glycogen content among the larvae and prepupae in different treatments, a 2-way ANOVA was performed. All data are expressed as means of triplicates with their standard errors. All analysis was performed on SPSS ver. 26 (IBM, Chicago, IL, USA). Principal component analysis (PCA) was performed by OriginLab 9 software (OriginLab, Northampton, MA, USA). Measured parameters for the biochemical composition, amino acid and fatty acid profiles were subjected to PCA to inspect the overall effect of spent coffee, donut dough or their blend. The standardized scores of the first two components, which explained the highest variation, were applied to prepare the biplots.

## 3. Results

### 3.1. Larval Productivity 

Both the BSF larvae and prepupae fed spent coffee were significantly shorter (*p* < 0.05) compared to the blend and dough treatments (Table 4). Meanwhile, the BSF larvae and prepupae were significantly heavier (*p* < 0.05) than the other treatments (Table 4). There was no significant interaction between food and stage on BSF larvae and prepupae lengths or weights (*p* > 0.05) (Table 4). There were significantly more prepupae in the dough treatment compared to all others. The lowest amount of prepupae were in the spent coffee treatment (*p* < 0.05) (Figure 1; Table 5).

The gross and net production of BSFL was significantly higher (*p* < 0.05) in the blend treatment compared to the others (Table 3). The final survival was significantly higher (*p* < 0.05) in the blend treatment, compared to the others, followed by the spent coffee treatment (Table 5).

### 3.2. Biochemical Composition

Moisture content was unaffected by treatments or stage (*p* > 0.05) (Figure 2); however, the other parameters were significantly affected. The protein content was significantly higher (*p* < 0.05) in prepupae fed dough compared to all others, while the lowest protein content was in the blend treatment for larvae, which was significantly lower (*p* < 0.05) than both the larvae and prepupae in the dough treatment (Figure 2). Lipid content was significantly higher (*p* < 0.05) in the prepupae fed the blend or dough as well as larvae fed the dough compared to all others (Figure 2). Finally, the glycogen content of larvae fed the dough was significantly higher (*p* < 0.05) than all others, except for prepupae in the dough treatment (Figure 2).

The two-way ANOVA showed that stage had a significant effect (*p* < 0.05) on the moisture, protein, lipid and glycogen content, while the food type had a significant effect (*p* < 0.05) on the protein, lipid and glycogen content. There was a significant interaction (*p* < 0.05) of food and stage on the glycogen content (Table 6).

### 3.3. Fatty Acid and Amino Acid Composition

With the exception of C2 (acetic acid), all the short chain fatty acids (C1–C8) were significantly higher (*p* < 0.05) in BSFL fed spent coffee compared to the blend or dough treatments (Table 7). Among the saturated fatty acids, the blend treatment led to significantly higher (*p* < 0.05) C14, C15, and C22 than the others while for the remaining SFA, these were significantly higher (*p* < 0.05) in BSFL fed spent coffee compared to the dough treatment (Table 7). For the monounsaturated fatty acids, these were either significantly higher or similar in BSFL fed spent coffee compared to the dough treatment. Among the polyunsaturated fatty acids (PUFA), C18:2n-6 were similar between the spent coffee and dough treatments (*p* > 0.05) but was significantly less (*p* < 0.05) compared to BSFL fed the blend. Although C18:3n-3 was significantly higher (*p* < 0.05) in the dough treatment, the long chain PUFA (LC-PUA) that included 20:4n-6 and C20:5n-3 were significantly higher (*p* < 0.05) in the spent coffee treatment compared to the others. No significant effect was observed for C22:6n-3 (*p* > 0.05) (Table 7).

Among the essential amino acids (EAA), histidine, leucine, methionine, phenylalanine and threonine were significantly higher (*p* < 0.05) in BSFL fed dough compared to spent coffee (Table 8). Only tryptophan among the EAA that was significantly higher (*p* < 0.05) in BSFL fed spent coffee compared to dough. Generally, the EAA composition of BSFL in the blended treatment followed that of spent coffee. For the nonessential amino acids (NEAA), arginine, aspartic acid, serine and tyrosine in BSFL fed dough were significantly higher (*p* < 0.05) compared to those in the spent coffee treatment. In BSFL fed spent coffee, only alanine was significantly higher (*p* < 0.05) among the NEAA compared to the dough treatment (Table 8).

### 3.4. Frass Composition

The nitrogen content of frass from the blend treatment was significantly higher (*p* < 0.05) than all others. The phosphorus and potassium content of the frass was significantly lower and higher, respectively, in the spent coffee treatment than all other treatments. Meanwhile, the frass calcium content was significantly lower (*p* < 0.05) in the dough treatment compared to all others (Figure 3).

### 3.5. Principle Component Analysis

A two-dimensional PCA plot for biochemical composition depicts a clear separation of experimental groups, mainly along the first two components (PC1 and PC2), together elucidating almost 100% of data variability (Figure 4A). The prevalent PC1 component (85.2% of the data variance) clustered protein, glycogen and lipid with donut dough; whereas PC2 (14.8% of the data variance) clustered moisture content with spent coffee.

Likewise, for amino acid profile, the prevalent PC1 component (68.2% of the data variance) clustered proline, glycine, valine, tyrosine, isoleucine, aspartic acid, serine, methionine, leucine, phenylalanine and threonine with dough (Figure 4B); while PC2 (26.6% of the data variance) clustered glutamic acid, taurine, hydroxyproline, alanine and ornithine with coffee. The two components also revealed a clear separation for fatty acid composition among coffee, dough or their blend as illustrated in Figure 4C. The primary trend observed for biochemical composition, amino acid profile and fatty acid composition was a differentiation of dough along the first PCA axis with respect to the coffee or blend.

## 4. Discussion

The current study used relatively large areas for the culture of BSFL over 35 days, where the amount of overall production including daily pupation were measured. Among the tested substrates, BSFL fed dough were the longest and heaviest, while the smallest BSFL were cultured from spent coffee. It has been similarly reported that spent coffee may not support optimal growth of BSFL and one of the contributors was higher amounts of indigestible fiber [31,32]. Other factors may also include the harder texture that poorly absorbs water, nutrient deficiencies and trace amounts of deleterious compounds such as caffeine and tannins. However, the overall production and amount of pupation was significantly highest in the blend treatment, which was mostly driven by the large discrepancy in survival among the treatments. In fact, the lowest survival came from the dough at 24% despite the larvae being heavier and longer than the other treatments. Based on daily observations, it appears likely that the cause for lower productivity (survival or growth) in the spent coffee and dough treatments was these substrates tended to dry out, despite being daily sprayed with water and the room had a relative humidity of around 45%. However, when the mister system was turned on it was observed that the water tended to fall off the sides of the spent coffee and dough, whereas a blend of these substrates appeared to have a softer texture throughout. Indeed, by the end of the study, there were some areas of the remaining dough/frass that were completely hardened whereas others were moist and soft. This inconsistency likely led to the lower survival but better growth of BSFL in the dough treatment. This assumption is based on researchers emphasizing the importance of substrate moisture, with an optimal range of 50–70%, for adequate BSFL survival and development [38,39]. 

In terms of production, the blend treatment was the best in this study, with an overall gross and net productivity of 5.51 g/day/m^3^ and 4.42 g/day/m^3^, respectively. Blends of different ingredients have been shown to improve BSFL production [8], which is consistent with other organisms. Production values of BSFL are not often presented in other studies, but such values may be useful to the industry in order to estimate spaces necessary to obtain a certain output and thus a viable business model. It is important to note, however, that the production values in this study are likely a substantial underestimation because approximately 60–80% of the total culture area in each layer was not utilized for BSFL production. 

In terms of the nutritional profile, the dough tended to increase the protein, lipid and glycogen content of the BSFL and/or BSFP, which seems to largely reflect the protein and lipid contents of the initial substrates. PC analysis also confirmed that protein, lipid and glycogen content clustered with dough; possibly indicating a prominent beneficial effect of dough on biochemical composition of BSFL and/or BSFP. Interestingly, the stage of the larvae significantly affected the protein and lipid content, which were both higher in the prepupae stage. It is important to note, however, that the prepupae are at a nonfeeding stage and their protein and fat content diminishes over time as they develop into adults (Liu et al. 2019). Thus, a likely reason for the prepupae having a relatively high protein and lipid content in this study was because they were harvested/stored within 12 h.

In addition to quantity of protein and fat, the quality of these macronutrients in BSFL is also important when formulating the diets of terrestrial and aquatic animals. The amino acid composition was generally higher in BSFL fed dough compared to spent coffee, which was also apparent from the PCA cluster analysis. It could be speculated that the spent coffee may have also had some amino acids and fatty acids destroyed during the roasting and brewing process that utilize high temperatures compared with the uncooked dough. However, the tested amino acid content of spent coffee was often higher than dough; nevertheless, important exceptions were found. Methionine was similar between spent coffee and dough substrates while lysine was actually lower in spent coffee than dough. These are important differences because both methionine and lysine are often limiting amino acids in the diets of both terrestrial and aquatic animals. Other studies have shown that these limiting amino acids can be influenced by the substrate, but the differences were numerically small. For example, Spranghers et al. [17] found minor differences in the amino acid composition of BSFL fed chicken feed, vegetable waste, biogas digestate or restaurant waste. Both Lalander et al. [16] and Fischer and Romano [15] did find some significant differences in the BSFL fed different waste products, but the numerical differences were also minor. The relatively stable amino acid profile of BSFL is certainly a benefit for the livestock industry. This is because despite the relatively low protein content of the tested substrates, the essential amino acid content of BSFL fed dough is comparable with soybeans [40], which is processed into soybean meal that often constitutes the main dietary protein source for many farmed terrestrial and aquatic animals [18].

In contrast to the dough treatment, the fatty acids were typically higher in BSFL fed spent coffee, followed by the blend treatment. This finding was especially pronounced for the short chain fatty acids (SCFA; C1–C8) as well as for palmitic acid (C16), stearic acid (C18), palmitoleic acid (C16:1n-7) and the long chain polyunsaturated fatty acids (LC-PUFA), that included arachidonic acid (20:4n-6; ARA) and eicosapentaenoic acid (20:5n-3; EPA). Studies have shown that SCFA can impart various health benefits to terrestrial and aquatic animals, and the origin of SCFA is from bacterial fermentation of indigestible carbohydrates [41,42]. Thus, the higher SCFA in the spent coffee treatment was likely due to the higher cellulose content that was fermented by bacteria inside BSFL, but more research is required to quantify this ability. In the case of LC-PUFA, these are required by some aquatic animals during their culture, especially marine animals. It is unlikely that the BSFL obtained LC-PUFA from the substrate and may indicate some ability for their synthesis. Using labelled fatty acids, Hoc et al. [43] found that BSFL could not synthesize polyunsaturated fatty acids (PUFA) but did emphasize that the enzymes responsible for PUFA synthesis should be investigated. Indeed, there are reports that when using substrates that likely lacked EPA and DHA, these were detected [15,44,45] whereas others found none [11,12,13,14]. Some PUFA and LC-PUFA have a wide range of health benefits for humans and therefore this should prompt additional studies. Finally, lauric acid was a major fatty acid in BSFL, which is consistent with other studies [46,47]. However, lauric acid was significantly lower in spent coffee treatment compared to those in the blend or dough treatments. Others similarly found that lauric acid in BSFL was lower in substrates with a higher content of indigestible fibres [17,47], which may explain this finding. Lauric acid does have some anti-microbial and anti-obesity properties to animals [48,49] that could be especially beneficial for human nutrition.

After harvesting the BSFL and BSFP, this leaves behind a substantial amount of frass that primarily consists of larvae excrement as well as dead larvae, chitin and any unconsumed substrate. It can be expected that with production of BSFL expanding, there will be increasing interest in the applications of frass, such as for crop farming [25,27,28]. Indeed, currently there are several commercially available BSFL frass products as organic fertilizers and soil amendments. Their value is largely based on having a similar or even higher nitrogen-phosphorous-potassium (NPK) content to other organic fertilizers [50]. For example, NPK from worm castings (1.5–2.5–1.3), compost from leaves (1.5–0.5–1.0) and poultry manure (1.5–1.0–0.5) [51] were less nitrogen heavy compared to the BSFL frass in this study at 4.2–0.31–0.63. However, there is the potential to alter the NPK content based on the provided food as well as amending with additional carbon. A more balanced NPK of BSFL frass at 4.4–5.2–4.1 was obtained when using a standard fly diet (50% wheat bran, 30% alfalfa meal and 20% corn meal) [26]. Additionally, elevating the carbon ratio to nitrogen ratio to 15, via sawdust additions, enhanced N and P retention as well as reduced the time necessary for composting [52]. In another study, Beesigamukama et al. [53] found that adding 10% biochar increased the K content and improved seed germination. Other essential nutrients, such as magnesium, iron, nickel, boron, manganese and zinc should also be measured in BSFL frass due to their importance for plant growth. Moreover, when these essential minerals fortify the plants they, in turn, can be more nutritious for human consumers [54]. Finally, it should be noted that another potential benefit to BSFL frass includes the chitin content, which has been shown to reduce pests and improve plant growth and health [55]. These areas of research should receive increasing attention to improve farming sustainability.

## 5. Conclusions

The use of readily available and low-cost BSFL food in the form of spent coffee grounds and donut dough can produce a sustainable ingredient for farmed terrestrial and aquatic animals. In fact, the BSFL fed dough had a comparable protein and amino acid content with soybeans. Dough was the prevailing factor influencing the amino acid profile and proximate composition. Nevertheless, BSFL fed spent coffee had small amounts of LC-PUFA that are essential for some commercially important fish/shellfish species. The use of a blend did appear to provide a more nutritionally balanced BSFL between using spent coffee and dough in terms of the SCFA and lauric acid content as well as substantially improving production. Based on visual observations, this may have been due to the blend having a softer texture likely due to drying out less compared to the other substrates. After harvesting, the left behind frass could have important applications as an organic fertilizer. Therefore, BSFL farming can potentially yield two resources to enhance both animal and plant farming in a sustainable manner.

## Figures and Tables

**Figure 1 insects-12-00332-f001:**
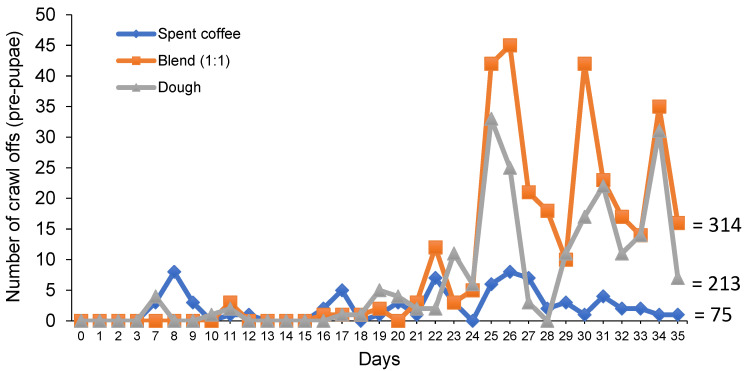
Number of black soldier fly prepupae collected each day over 35 days when the larvae were cultured on spent coffee, dough or an equal blend of these ingredients.

**Figure 2 insects-12-00332-f002:**
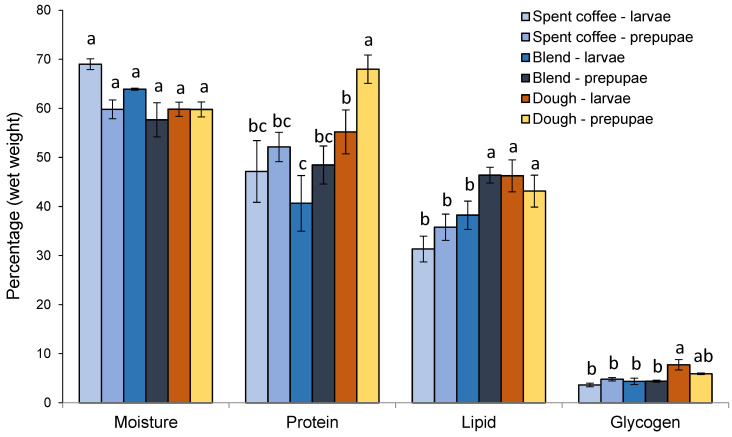
Mean (±SE) moisture, protein, lipid and glycogen (% dry weight) of black soldier fly larvae and prepupae when cultured with spent coffee, dough or an equal blend of these after 35 days. Different letters indicate significant differences (*p* < 0.05).

**Figure 3 insects-12-00332-f003:**
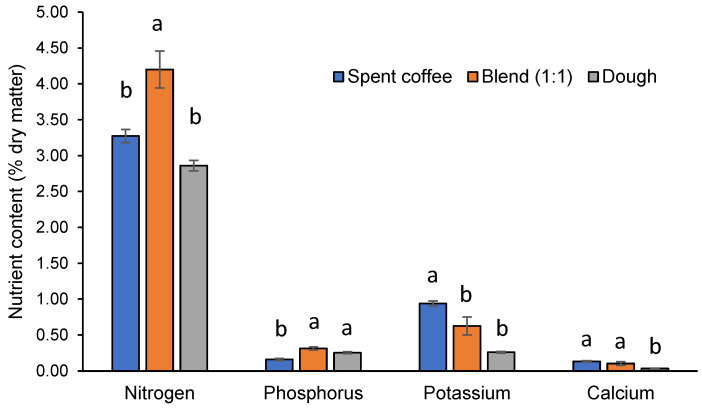
Mean (±SE) nitrogen, phosphorus, potassium and calcium (% dry matter) of frass from black soldier fly larvae when cultured with spent coffee, dough or an equal blend of these after 35 days. Different letters indicate significant differences (*p* < 0.05).

**Figure 4 insects-12-00332-f004:**
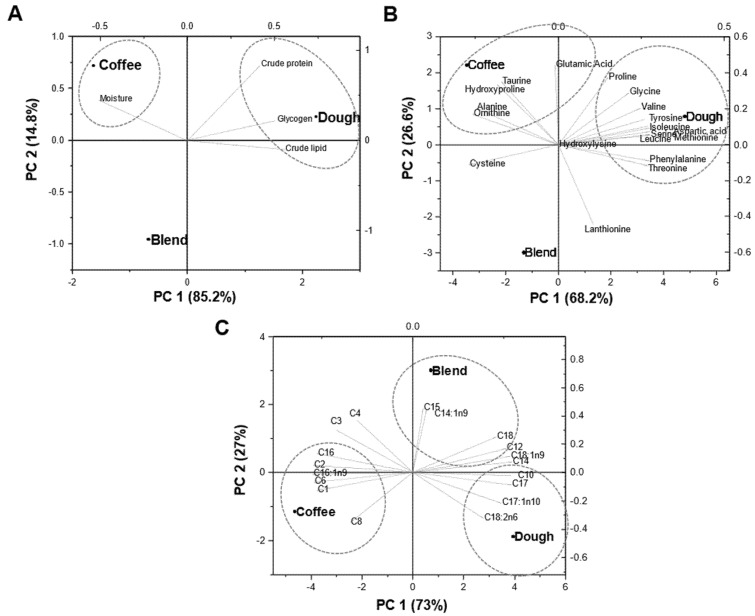
Principal Component Analysis (PCA) representing the contribution of (**A**) biochemical composition, (**B**) amino acid composition and (**C**) fatty acid investigated. The variable coordination is presented by the complementary cases analysis showing distribution of spent coffee, donut dough or their blend in the (PC1 × PC2) coordination plane.

**Table 1 insects-12-00332-t001:** Biochemical composition of the spent coffee and dough (% on “as is” basis).

Biochemical Composition	Spent Coffee	Dough
Moisture	69.42	42.40
Crude protein	4.80	10.29
Crude lipid	5.14	8.71
Cellulose	4.61	0.83
Ash	0.47	0.91
Potassium	0.14	0.11
Phosphorus	0.03	0.07

**Table 2 insects-12-00332-t002:** Amino acid composition (g/100 g on dry weight basis) of the spent coffee grounds, dough or equal blend of these ingredients.

Amino Acids	Spent Coffee	Blend (1:1)	Dough
Essential amino acids			
Histidine	0.18	0.19	0.18
Isoleucine	0.48	0.38	0.32
Leucine	1.04	0.74	0.57
Lysine	0.13	0.22	0.23
Methionine	0.13	0.12	0.13
Phenylalanine	0.67	0.48	0.40
Threonine	0.18	0.22	0.22
Tryptophan	0.08	0.08	0.10
Valine	0.69	0.48	0.36
Nonessential amino acids			
Alanine	0.53	0.37	0.26
Arginine	0.02	0.24	0.33
Aspartic acid	0.77	0.49	0.36
Cysteine	0.02	0.16	0.20
Glutamic acid	1.82	2.49	2.73
Glycine	0.67	0.43	0.31
Serine	0.08	0.29	0.36
Proline	0.53	0.75	0.84
Taurine	0.13	0.20	0.21
Tyrosine	0.32	0.24	0.16

**Table 3 insects-12-00332-t003:** Fatty acid composition (%) of the spent coffee grounds, dough or equal blend of these ingredients.

Fatty Acids	Spent Coffee	Blend (1:1)	Dough
Saturated fatty acids			
C14	0.08	0.34	0.56
C15	0.02	0.04	0.04
C16	33.74	31.12	28.77
C17	0.11	0.10	0.08
C18	7.42	5.22	3.33
C20	2.77	1.52	0.42
C22	0.63	0.39	0.17
C24	0.23	0.18	0.11
Monounsaturated fatty acids			
C16:1n9	0.04	0.12	0.19
C18:1n9	8.87	27.35	43.79
C20:1n9	0.34	0.47	0.53
Polyunsaturated fatty acids			
C18:2n6	43.17	29.71	17.91
C18:3n3	1.38	1.38	1.43
C20:3n3	0.09	0.06	0.00
C20:5n3	0.00	0.02	0.00
C22:6n3	0.00	0.00	0.00

**Table 4 insects-12-00332-t004:** Mean (±SE) lengths (cm) and weights (g) of black soldier fly larvae and prepupae when cultured with spent coffee, dough or an equal blend of these after 35 days. Different superscripted letters in each column indicate significant differences (*p* < 0.05).

Treatment	Larval Length	Prepupae Length	Larval Weight	Prepupae Weight
Spent coffee	16.86 ± 0.29 ^b^	17.26 ± 0.42 ^b^	0.11 ± 0.01 ^c^	0.11 ± 0.01 ^c^
Blend (1:1)	19.12 ± 0.72 ^a^	20.60 ± 0.36 ^a^	0.18 ± 0.01 ^b^	0.19 ± 0.02 ^b^
Dough	21.44 ± 0.59 ^a^	21.57 ± 0.60 ^a^	0.23 ± 0.01 ^a^	0.23 ± 0.01 ^a^
	Length		Weight	
Main effects *	F	Sig	F	Sig
Food type	36.71	0.001	89.47	0.001
Stage	2.44	0.144	0.364	0.558
Food × stage	0.935	0.419	0.159	0.855

* Food type refers to spent coffee, blend or dough; stage refers to larvae versus prepupae.

**Table 5 insects-12-00332-t005:** Survival and productivity (mean ± SE) of black soldier fly larvae and prepupae when cultured with spent coffee, dough or an equal blend of these after 35 days. Different superscripted letters in each row indicate significant differences (*p* < 0.05).

Production	Spent Coffee	Blend	Dough
Gross production (g/day/m^3^)	1.83 ± 0.58 ^b^	5.51 ± 1.01 ^a^	1.68 ± 1.00 ^b^
Net production (g/day/m^3^)	0.75 ± 0.58 ^b^	4.42 ± 1.02 ^a^	0.60 ± 1.01 ^b^
Survival (%)	45.13 ± 10.2 ^b^	81.16 ± 12.5 ^a^	24.56 ± 4.76 ^c^
Total larvae (g)	42.32 ± 13.61 ^b^	127.42 ± 23.57 ^a^	38.95 ± 23.32 ^b^
Total prepupae (g)	2.56 ± 0.84 ^b^	20.70 ± 3.29 ^a^	15.40 ± 3.68 ^a^
Total production (g)	44.89 ± 13.1 ^b^	148.12 ± 20.44 ^a^	54.35 ± 20.99 ^b^

**Table 6 insects-12-00332-t006:** Main effects of food type and stage on the moisture, protein, lipid and glycogen content of black soldier fly larvae and prepupae after 35 days.

Main Effects *	Moisture		Protein		Lipid		Glycogen	
	F	Sig	F	Sig	F	Sig	F	Sig
Food type	3.26	0.74	7.38	0.01	8.93	0.01	13.58	0.01
Stage	11.11	0.01	5.28	0.04	1.94	0.18	0.21	0.65
Food × stage	3.04	0.08	0.37	0.69	2.14	0.16	3.69	0.06

* Food type refers to spent coffee, blend or dough; stage refers to larvae versus prepupae.

**Table 7 insects-12-00332-t007:** Fatty acid composition (mg/g “as is basis”) (±SE) of black soldier fly larvae when cultured with spent coffee, dough or an equal blend of these after 35 days. Different superscripted letters in each row indicate significant differences (*p* < 0.05).

Fatty Acids	Spent Coffee	Blend	Donut Dough
Short chain fatty acids			
C1	0.25 ± 0.03 ^a^	0.16 ± 0.01 ^b^	0.15 ± 0.03 ^b^
C2	5.61 ± 0.22 ^a^	3.48 ± 0.42 ^ab^	1.61 ± 0.34 ^b^
C3	0.16 ± 0.01 ^a^	0.15 ± 0.03 ^a^	0.04 ± 0.01 ^b^
C4	0.33 ± 0.07 ^a^	0.35 ± 0.22 ^a^	0.09 ± 0.01 ^b^
C6	0.15 ± 0.02 ^a^	0.10 ± 0.01 ^b^	0.09 ± 0.01 ^b^
C8	0.38 ± 0.04 ^a^	0.23 ± 0.01 ^b^	0.30 ± 0.05 ^a^
Saturated fatty acids			
C10	2.19 ± 0.18 ^c^	3.82 ± 0.19 ^b^	4.59 ± 0.23 ^a^
C12	66.44 ± 6.44 ^b^	158.9 ± 6.52 ^a^	158.8 ± 9.74 ^a^
C14	11.39 ± 0.87 ^c^	24.40 ± 1.13 ^a^	19.77 ± 1.21 ^b^
C15	0.22 ± 0.03 ^b^	0.60 ± 0.08 ^a^	0.34 ± 0.03 ^b^
C16	62.99 ± 3.99 ^a^	38.35 ± 3.27 ^b^	30.77 ± 2.00 ^b^
C17	0.64 ± 0.03 ^a^	0.57 ± 0.04 ^a^	0.41 ± 0.05 ^b^
C18	7.69 ± 0.21 ^a^	5.24 ± 0.46 ^b^	4.04 ± 0.41 ^b^
C20	0.63 ± 0.01 ^a^	0.25 ± 0.03 ^b^	0.03 ± 0.03 ^c^
C21	4.98 ± 0.27 ^a^	3.90 ± 0.41 ^b^	0.00 ± 0.00 ^c^
C22	0.13 ± 0.00 ^b^	0.36 ± 0.06 ^a^	0.04 ± 0.00 ^c^
C24	0.08 ± 0.00 ^a^	0.03 ± 0.00 ^b^	0.00 ± 0.00 ^c^
Monounsaturated fatty acids			
C14:1n5	0.45 ± 0.02 ^a^	0.57 ± 0.08 ^a^	0.17 ± 0.02 ^b^
C15:1n5	0.05 ± 0.00 ^a^	0.08 ± 0.00 ^a^	0.02 ± 0.00 ^a^
C16:1n7	9.09 ± 0.50 ^a^	9.35 ± 0.26 ^a^	1.10 ± 0.14 ^b^
C17:1n10	1.45 ± 0.10 ^a^	0.75 ± 0.12 ^b^	1.51 ± 0.20 ^a^
C18:1n9	26.57 ± 1.22 ^a^	27.24 ± 2.03 ^a^	25.08 ± 2.17 ^a^
C20:1n9	0.26 ± 0.02 ^a^	0.16 ± 0.01 ^b^	0.00 ± 0.00 ^c^
C22:1n9	0.10 ± 0.00 ^a^	0.11 ± 0.01 ^a^	0.03 ± 0.00 ^a^
Polyunsaturated fatty acids			
C18:2n6	64.38 ± 3.44 ^a^	29.11 ± 3.01 ^b^	52.87 ± 6.11 ^a^
C18:3n3	1.69 ± 0.06 ^b^	2.21 ± 0.29 ^b^	5.92 ± 0.25 ^a^
C20:4n6	0.28 ± 0.04 ^a^	0.02 ± 0.01 ^b^	0.00 ± 0.00 ^b^
C20:5n3	0.13 ± 0.00 ^a^	0.00 ± 0.00 ^b^	0.00 ± 0.00 ^b^
C22:6n3	0.02 ± 0.00 ^a^	0.00 ± 0.00 ^a^	0.00 ± 0.00 ^a^

**Table 8 insects-12-00332-t008:** Amino acid composition (g/100 g on “as is basis”) (±SE) of black soldier fly larvae when cultured with spent coffee, dough or an equal blend of these after 35 days. Different superscripted letters in each row indicate significant differences (*p* < 0.05).

	Spent Coffee	Blend	Donut Dough	Soybean *
Essential amino acids				
Histidine	1.21 ± 0.03 ^b^	1.24 ± 0.02 ^b^	1.44 ± 0.04 ^a^	1.22
Isoleucine	1.98 ± 0.04 ^a^	1.99 ± 0.02 ^a^	2.12 ± 0.01 ^a^	2.10
Leucine	2.85 ± 0.08 ^b^	2.87 ± 0.04 ^b^	3.14 ± 0.05 ^a^	3.57
Lysine	2.47 ± 0.04 ^ab^	2.32 ± 0.09 ^b^	2.60 ± 0.05 ^a^	2.99
Methionine	0.70 ± 0.02 ^b^	0.71 ± 0.01 ^b^	0.81 ± 0.01 ^a^	0.68
Phenylalanine	1.67 ± 0.04 ^b^	1.77 ± 0.02 ^b^	1.91 ± 0.03 ^a^	2.33
Threonine	1.50 ± 0.05 ^b^	1.61 ± 0.03 ^ab^	1.72 ± 0.02 ^a^	1.85
Tryptophan	0.62 ± 0.02 ^a^	0.49 ± 0.02 ^b^	0.48 ± 0.03 ^b^	0.65
Valine	3.01 ± 0.04 ^a^	2.96 ± 0.05 ^a^	3.18 ± 0.09 ^a^	2.26
Nonessential amino acids				
Alanine	3.25 ± 0.05 ^a^	2.97 ± 0.08 ^b^	2.84 ± 0.08 ^b^	2.02
Arginine	1.55 ± 0.09 ^b^	1.66 ± 0.05 ^b^	2.18 ± 0.04 ^a^	3.43
Aspartic acid	3.08 ± 0.12 ^b^	3.18 ± 0.08 ^b^	3.77 ± 0.05 ^a^	5.42
Cysteine	0.29 ± 0.00 ^a^	0.29 ± 0.00 ^a^	0.28 ± 0.00 ^a^	0.73
Glutamic acid	3.95 ± 0.04 ^a^	3.59 ± 0.09 ^b^	3.84 ± 0.03 ^a^	8.58
Glycine	2.59 ± 0.05 ^a^	2.46 ± 0.04 ^a^	2.75 ± 0.16 ^a^	1.99
Serine	1.41 ± 0.05 ^b^	1.45 ± 0.02 ^b^	1.70 ± 0.10 ^a^	2.32
Proline	2.39 ± 0.03 ^a^	2.33 ± 0.01 ^a^	2.41 ± 0.09 ^a^	2.34
Taurine	0.07 ± 0.00 ^a^	0.06 ± 0.00 ^a^	0.07 ± 0.00 ^a^	0.00
Tyrosine	2.28 ± 0.11 ^b^	2.29 ± 0.06 ^b^	2.70 ± 0.04 ^a^	0.40

* US Soybeans (https://ussec.org/wp-content/uploads/2015/10/US-Soybean-Meal-Information.pdf).

## Data Availability

Data is available upon reasonable request.

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
