# Peer review of "Conversion of Spent Coffee and Donuts by Black Soldier Fly (Hermetia illucens) Larvae into Potential Resources for Animal and Plant Farming"

_insects, 2021, doi:10.3390/insects12040332_

Round 1

Reviewer 1 Report

The article provides an elegant study on growth and nutritional composition of black soldier fly larvae on common waste products in the US. However, the experiment is unrepeated and (seemingly) unrandomized/unreplicated experiment, and has flaws.

I have the following major comments that would need to be addressed before the article could be published.

  • This study is not repeated in time. Furthermore, how the three treatments were distributed to the devices (each with three layers is unclear. I suspect that each device dealt with 1 treatment, and as such there was no randomization of this unrepeated experiment? Therefore, the study cannot be published.
  • Furthermore, the authors hint in the Discussion on some flaws, that would make a repeat essential. For instance, there is the issue with drying out of the substrate, and the statement that ‘60%-80% of the total culture area in each layer was not utilized’. The blend seemsed to have dried out, requiring the study to be repeated.
  • Why were pupae not measured for the amino acid and fatty acid composition?
  • Please fix the title into a scientific statement. This is a scientific publication, not a popular blog.
  • Please fix the Simple Summary. There are numerous spelling and grammatical mistakes (illustrating authors did not bother to proofread): the acronym BSFL is not introduced; ‘solider’ fly; ‘production can produce’; etc.
  • Introduction: there is a need to properly introduce the use of black soldier fly as animal feed (which is also partly the reason why chemical composition of larvae and pupae was conducted?).
  • Introduction: some essential literature on black soldier fly frass (its composition as a function of substrate, its use) may be missing, and the statement that ‘Nevertheless, there is limited information on the effects of substrates on the nutrient composition of BSFL frass’ is misleading. I suggest the authors update their literature search? (e.g. refer to all 2020 Beesigamukama et al papers: Waste Management (2020), 119, 183-194; Agronomy (2020) 10, 1395; Plos One (2020) 15, e0238154; Frontiers in Plant Science (2020), 11). 84-87: The discussion on applicability of C/N ratio (as a substrate) is missing.
  • Discussion: also in the Discussion (294-302), authors did not refer whatsoever to other literature. There are the many other studies done on nutritional composition of black soldier flies! The presence of lysine and methionine, as limiting amino-acids, has been reported before (several times). 326-328: Existing literature on NPK values and value of frass as fertilizer (see above) is missing.
  • Discussion: can the authors (try to) explain why larval survival was lower yet growth faster (heavier larvae) on dough?

Other comments:

  • 26-27: ‘compared’ x 2.
  • 33: should be donut dough.
  • 64: dot missing.
  • 136: brackets missing.
  • 137: subject in sentence missing.
  • 175: Not proper English
  • 235: Please provide a proper reference. Also the fact that soybean is the main animal feed ingredient belongs in the Introduction.  
  • 270: misspelled.
  • 322: Please list those.
  • Figure 1 is not needed.

Author Response

Reviewer comment #1:  This study is not repeated in time. Furthermore, how the three treatments were distributed to the devices (each with three layers is unclear. I suspect that each device dealt with 1 treatment, and as such there was no randomization of this unrepeated experiment? Therefore, the study cannot be published.

Response to comment #1:  So, this study was done in triplicate and therefore it was not necessary to replicate through time.  Each device had 3 layers – yes – but each device (of which there were 3 devices/treatment) was a replicate (that had 3 layers).  In other words, each treatment was triplicated, and each replicate had 3 layers (which is a rather robust study).

Reviewer comment #2:  Furthermore, the authors hint in the Discussion on some flaws, that would make a repeat essential. For instance, there is the issue with drying out of the substrate, and the statement that ‘60%-80% of the total culture area in each layer was not utilized’. The blend seemsed to have dried out, requiring the study to be repeated.

Response to comment #2:  So the dough tended to dry out – yes.  Not sure how replicating this through time would alter the tendency of dough to dry.  Especially considering the relative humidity was relatively high and we routinely sprayed the substrates each day (as stated).

Reviewer comment #3:   Why were pupae not measured for the amino acid and fatty acid composition?

Response to comment #3:  after measuring the proximate composition we ran out of sample for such analysis.

Reviewer comment #4:  Please fix the title into a scientific statement. This is a scientific publication, not a popular blog.

Response to comment #4:  thank you – we have now changed.

Reviewer comment #5:  Please fix the Simple Summary. There are numerous spelling and grammatical mistakes (illustrating authors did not bother to proofread): the acronym BSFL is not introduced; ‘solider’ fly; ‘production can produce’; etc.

Response to comment #5: Thank you for pointing this out – we have fixed these grammatical errors.

Reviewer comment #6:  Introduction: there is a need to properly introduce the use of black soldier fly as animal feed (which is also partly the reason why chemical composition of larvae and pupae was conducted?).

Response to comment #6:  Yes, thank you, we have included some additional sentences on this.

Reviewer comment #7:  Introduction: some essential literature on black soldier fly frass (its composition as a function of substrate, its use) may be missing, and the statement that ‘Nevertheless, there is limited information on the effects of substrates on the nutrient composition of BSFL frass’ is misleading. I suggest the authors update their literature search? (e.g. refer to all 2020 Beesigamukama et al papers: Waste Management (2020), 119, 183-194; Agronomy (2020) 10, 1395; Plos One (2020) 15, e0238154; Frontiers in Plant Science (2020), 11). 84-87: The discussion on applicability of C/N ratio (as a substrate) is missing.

Response to comment #7:  Thank you very much for these valuable references!  We have included all in this manuscript and are very helpful, interesting and enlightening. 

Reviewer comment #8:  Discussion: also in the Discussion (294-302), authors did not refer whatsoever to other literature. There are the many other studies done on nutritional composition of black soldier flies! The presence of lysine and methionine, as limiting amino-acids, has been reported before (several times). 326-328: Existing literature on NPK values and value of frass as fertilizer (see above) is missing.

Reviewer comment #8:  Thank you for pointing this out – yes, we should have discussed more on these important areas.  We have included additional sentences in the discussion that addresses these.

Reviewer comment #9:  Discussion: can the authors (try to) explain why larval survival was lower yet growth faster (heavier larvae) on dough?

Response to comment #9:  We have included two additional sentences based mostly on what we observed and believe was the cause.

Reviewer comments #10:  Other comments:

  • 26-27: ‘compared’ x 2.
  • 33: should be donut dough.
  • 64: dot missing.
  • 136: brackets missing.
  • 137: subject in sentence missing.
  • 175: Not proper English
  • 235: Please provide a proper reference. Also the fact that soybean is the main animal feed ingredient belongs in the Introduction.  
  • 270: misspelled.
  • 322: Please list those.
  • Figure 1 is not needed.

Response to comments #10:  We have adopted all above, except that a “proper” reference regarding the nutritional value of soybean meal that is included as a footnote is in fact valid.  This is a recent reference, comprehensive in its analysis as well as including analysis from different areas of the world.  Moreover, the analysis done was at the same place as this study (Agricultural Experiment Station Chemical Laboratories (AESCL) at University of Missouri-Columbia), and thus the results are reliable.  Additionally, we would like to keep Figure 1 because it can provide useful information to farmers in terms of when to expect to see prepupae and their frequency, so they may manage breeding and harvesting.

Reviewer 2 Report

Thank you for the opportunity to review “Black soldier fly (Hermetia illucens) larvae production: Converting an unbalanced breakfast of coffee and donuts into high quality resources for animal and plant farming” submitted to the journal Insects. This research paper is very timely and relevant in the current context of black soldier fly rearing and utility. The paper is well-written and contribute novel information to the field. In addition to the line edits suggested below, there are two additional items for consideration that I would like to see addressed in the revised version of the manuscript:

  1. I would like to see the authors provide a nutritional analysis of the raw ingredients used in the experiment (spent coffee grounds and donut dough). Given the level of detail provided in the analysis of the larve and prepupa, it would support a better understanding of the conversion of feedstock into insect and the residual in the frass.
  2. Some sort of multivariate analysis should be performed to examine the relationship that confounds the nutritional parameters that are measured (protein, lipid, glycogen) (fatty acids)(amino acids).

In the discussion, I would like to see more text related to the application of the frass to plant farming. This is emphasized in the title but is overlooked in the presentation of the results and discussion.

Author Response

Thank you for the opportunity to review “Black soldier fly (Hermetia illucens) larvae production: Converting an unbalanced breakfast of coffee and donuts into high quality resources for animal and plant farming” submitted to the journal Insects. This research paper is very timely and relevant in the current context of black soldier fly rearing and utility. The paper is well-written and contribute novel information to the field. In addition to the line edits suggested below, there are two additional items for consideration that I would like to see addressed in the revised version of the manuscript:

Reviewer comment #1:  I would like to see the authors provide a nutritional analysis of the raw ingredients used in the experiment (spent coffee grounds and donut dough). Given the level of detail provided in the analysis of the larve and prepupa, it would support a better understanding of the conversion of feedstock into insect and the residual in the frass.

Response to comment #1:  We sent out the ingredients for amio acid and fatty acid composition – these are shown in Table 2 and 3.

Reviewer comment #2:  Some sort of multivariate analysis should be performed to examine the relationship that confounds the nutritional parameters that are measured (protein, lipid, glycogen) (fatty acids)(amino acids).

Response to comment #2:  We have performed principle component analysis (PCA) to illustrate the clustering/relationship of the treatments to the biochemical composition, which is provided as a new figure.

Reviewer comment #3:  In the discussion, I would like to see more text related to the application of the frass to plant farming. This is emphasized in the title but is overlooked in the presentation of the results and discussion.

Response to comment #3:  Thank you – we have added more information on this in the discussion.

Reviewer 3 Report

Authors investigated BSF larval production for converting the spent coffee and donuts. Although the study could be of interest to the journal, the paper has to many flaws to be considered for publication.
  Title: authors wrote "an unbalanced breakfast of coffee and donuts" in the title, but the article is for the spent coffee and donuts (not breakfast of coffee). Please modify it for more exact. Line 120, what is their unit, mg/g, %, or others? The data is average? what is their error? Line 136, it should be "(final weight – initial weight)". Line 177 Table 2, please clarify "stage" and "Food * Stage" in Table 2 and 4. Line 181, "mix" is blend in Fig. 1, keep the same name across the article. It looks the Fig.1 is not shown completely(=31??, =21??, =75??). Line 184, is there some significant differences in Fig 2a? Please mark them with "abcd". Line 252, I suggest the authors add some discussion for larval production comparing to those converting other organic waste, and evaluate it. Thus the audience could know more information for this application.        

Author Response

Authors investigated BSF larval production for converting the spent coffee and donuts. Although the study could be of interest to the journal, the paper has to many flaws to be considered for publication.

Reviewer comment #1: Title: authors wrote "an unbalanced breakfast of coffee and donuts" in the title, but the article is for the spent coffee and donuts (not breakfast of coffee). Please modify it for more exact. 

Response to comment #1:  This has been changed.

Reviewer comment #2:  Line 120, what is their unit, mg/g, %, or others? The data is average? what is their error? 

Response to comment #2:  This was in %.  The values were not done in duplicate so the error cannot be provided.  We were not intending to do statistics among the substrates, but rather to just provide some background information regarding their biochemical composition.  Also, in our experience with the analysis of these biochemical parameters, when paying for triplicate samples in the past the values are nearly identical (as opposed to samples of the experimental animals/insects).

Reviewer comment #3:  Line 136, it should be "(final weight – initial weight)".

Response to comment #3:  Thank you for pointing this out.  We have changed.

Reviewer comment #4: Line 177 Table 2, please clarify "stage" and "Food * Stage" in Table 2 and 4.

Response to comment #4:  this has been clarified in a new footnote.

Reviewer comment #5: Line 181, "mix" is blend in Fig. 1, keep the same name across the article. It looks the Fig.1 is not shown completely(=31??, =21??, =75??).

Response to comment #5:  Thank you for pointing this out -- we have fixed these.

Reviewer comment #6: Line 184, is there some significant differences in Fig 2a? Please mark them with "abcd".

Response to comment #6:  we have now included “a” s throughout to indicate there were no significant differences.

Reviewer comment #7:  Line 252, I suggest the authors add some discussion for larval production comparing to those converting other organic waste, and evaluate it. Thus the audience could know more information for this application.        

Response to comment #7:  We have done several literature searches on larval production in order to make this comparison, but have not come across any.  We would certainly be grateful to any that could be provided if we missed some.

Round 2

Reviewer 3 Report

The paper still need a minor revision to be considered for publication.

For the title, I would use "potential resources" instead of "high quality resources". It still need more work to evaluate how the quality is high. please also revise it in Line 19, 21, 57, 454.

Line 101, Could you use "m"(meter) instead of "feet"? Also for Line 106-108.

Line 102, Please also add the "xx °C" after "88 °F". The "≈45%" should be revise to "∼45%" also for Line 347.

Line 227, Could modify the Fig. 1 to cumulative chart, which may be more clear to show the prepupa increasing tend. 

Line 254, Fig. 2a.

Line 456-457. PC analysis? Delete the "PC analysis also confirmed that", just write "Dough is the prevailing ..."

Author Response

The paper still need a minor revision to be considered for publication.

For the title, I would use "potential resources" instead of "high quality resources". It still need more work to evaluate how the quality is high. please also revise it in Line 19, 21, 57, 454.

Response:  This has been changed to say "potential resources" and this wording or similar wording in the text.

Line 101, Could you use "m"(meter) instead of "feet"? Also for Line 106-108.

Response:  We have now used meter instead of feet

Line 102, Please also add the "xx °C" after "88 °F". The "≈45%" should be revise to "∼45%" also for Line 347.

Response: We have included the Celcius  value and now said "around" instead of "≈"

Line 227, Could modify the Fig. 1 to cumulative chart, which may be more clear to show the prepupa increasing tend. 

Response:  we measured the amount of pupation on a daily basis and we feel that showing readers that the amounts that were self-harvested is variable and not necessarily a linear line.

Line 254, Fig. 2a.

Response:  We have fixed this (it was previously Fig. 1a, and now we fixed to Fig. 2a)

Line 456-457. PC analysis? Delete the "PC analysis also confirmed that", just write "Dough is the prevailing ..."

Response:  We have deleted "PC analysis also confirmed that"